# Prevalence and associated factors of occupational injuries in an industrial city in Ghana

Michael Tetteh Asiedu[1], Douglas Aninng Opoku[1,2]*, Nana Kwame Ayisi-Boateng[3,4], Joseph Osarfo[5], Alhassan Sulemana[6], Aliyu Mohammed[7], John Amissah[7], Jennifer Ashilevi[4], Ayongo Mate-Kole[8], Felix Agyemang Opoku[1], Isaac Kofi Yankson[9], Emmanuel Kweku Nakua[7]

1 Department of Occupational and Environmental Health, School of Public Health, Kwame Nkrumah University of Science and Technology, Kumasi, Ghana, 2 Allen Clinic, Family Healthcare Services, Kumasi, Ghana, 3 Department of Medicine, School of Medicine and Dentistry, Kwame Nkrumah University of Science and Technology, Kumasi, Ghana, 4 University Hospital, Kwame Nkrumah University of Science and Technology, Kumasi, Ghana, 5 Department of Community Health, School of Medicine, University of Health and Allied Health Science, Ho, Ghana, 6 Department of Environmental Science, College of Science, Kwame Nkrumah University of Science and Technology, Kumasi, Ghana, 7 Department of Epidemiology and Biostatistics, School of Public Health, Kwame Nkrumah University of Science and Technology, Kumasi, Ghana, 8 Family Medicine Sub BMC, Korle-Bu Teaching Hospital, Accra, Ghana, 9 Council for Scientific and Industrial Research-Building and Road Research Institute, Kumasi, Ghana

* douglasopokuaninng@gmail.com

**Data Availability Statement:** All relevant data are within the manuscript and its Supporting Information files.

## Abstract

### Background

Workers are exposed to workplace hazards which increase their risk of occupational injury. Data on occupational injuries and associated factors are important for planning and informing national policy regarding workplace health and safety. This study sought to estimate the prevalence and factors associated with occupational injuries among workers in an industrial city in Ghana.

### Methods

A community-based cross-sectional survey was conducted among 459 workers in the Tema industrial enclave in Ghana from 22nd December 2020 to 27th February 2021. Participants were recruited using a two-stage sampling technique. Eight communities were randomly selected from twenty-five communities in the first stage while households in each community were randomly selected in the second stage. Data on socio-demographic characteristics, occupational health and safety and occupational injuries were collected. Logistic regression was used to examine the relationship between occupational injuries and associated factors.

### Results

The mean age of the workers was 33.9 (±6.8) years with a range of 21–53 while over 18.1% of them were working at the Port and Harbour. The prevalence of occupational injury among

**Funding:** The authors received no specific funding for this work.

**Competing interests:** The authors have declared that no competing interests exist.

the workers in the preceding twelve months was 64.7%. The mechanism of injury was mainly the use of working tools (45.8%) and hot surfaces, substances or chemicals (14.1%). Being a casual staff (AOR: 2.26, 95%CI: 1.04–4.92), working at Port and Harbour (AOR: 3.77, 95%CI: 1.70–8.39), no health and safety training (AOR: 2.18, 95%CI: 1.08–4.39), dissatisfaction with health and safety measures (AOR: 4.31, 95%CI: 2.12–8.78) and tertiary education (AOR: 0.03, 95%CI: 0.01–0.10) were significantly associated with occupational injuries.

## Conclusion

The prevalence of occupational injuries in this study was high. Promoting machine tools' safety, health and safety training, and satisfaction with health and safety measures through rewarding workers who do not sustain injuries could be key to employees' health and safety.

## Introduction

Despite significant strides in the promotion of health and safety at the workplace, most industrial workers in low and middle-income countries (LMICs) work under unfavourable working conditions such as poor lighting, lack of personal protective equipment (PPE), inadequate working space, long working hours, and poor ventilation [1–3]. Exposures of workers to undesirable occupational hazards increase their susceptibility to occupational injuries and illnesses [3–7]. The global burden of occupational injuries and diseases is estimated to be 1.9 million deaths in 2019 [8]. The economic cost due to occupational injuries accounts for up to 6.0% of a nation's gross domestic product (GDP) with a global average of 4.0% [5, 6].

The risk of work-related injuries varies generally across the globe. The risk remains higher in LMICs. LMICs have an estimated 10 to 20 times increased risk of sustaining occupational injuries compared to their counterparts in developed countries [9]. The disparity in the distribution of these burdens could be attributed to inadequate attention given to occupational health and safety, particularly, in preventing work-related accidents, illness and injuries [10]. A systematic review by Alamneh *et al.* (2020) reported a pooled prevalence of occupational injuries of 44.7% in Ethiopia [11]. The construction industry is considered to have one of the most hazardous environments and a study in Uganda recorded 32.4% of occupational injuries [12]. The types of occupational injuries workers suffer include cuts, lacerations, fractures, open wounds and concussions [13, 14].

The occurrence of occupational injuries is influenced by several factors including socio-cultural, economic and individual factors [15–17]. The constraint-response accident model is one of the theories which has been widely used in the literature to explain the factors contributing to occupational injuries [18]. This theory recognizes the contributions of each worker at the workplace and predicts that while performing their responsibilities they are constrained by some factors and their reactions to these constraints generate another layer of constraints for other workers who rely on the former to take action [18, 19]. The interplay between management, organization and operational factors contributes to occupational injuries among workers. Proximal and distal factors are the two key concepts used to explain the occurrence of occupational injuries in this theory [18, 19]. The proximal factors are socio-demographic and individual characteristics (including age, work experience, adherence to PPE, exposure to occupational hazards etc) which directly influence the occurrence of occupational injuries

[18]. Conversely, distal factors are managerial and organizational factors (including income, number of working hours, commitment to health and safety measures etc.) which indirectly influence the occurrence of occupational injuries [18].

Safety training is one of the key strategies that can be used to reduce occupational injuries [20]. Previous studies conducted in Ethiopia and Uganda reported that workers who did not receive safety training had about three times increased risk of occupational injury [21–23]. Other factors known to increase the burden of occupational injuries include low educational background and work experience of less than two years [24]. Adherence to PPE usage also has the potential to reduce the incidence of occupational injuries [25]. Meanwhile, its use in LMICs is low due to inadequate supply [17]. Moreover, in an instance where it is supplied, timely changing becomes a challenge. Previous studies have reported failure to use PPE at the workplace is associated with three times increased risk of occupational injuries [22, 23].

Ghana is not spared of the burden of occupational injuries as it continues to claim the lives of many. Earlier studies in Ghana have estimated injury prevalence across several industries including 57.9% among general construction workers [16], 21.8% among solid waste collectors [26], and 29.7% among healthcare workers [27]. Apart from the few studies mentioned, little is known about the prevalence and factors associated with occupational injuries among workers in a major industrial city with over five hundred companies and the largest Port in Ghana. These studies only focused on estimating injury prevalence among specific working populations in Ghana [7, 13, 16, 26–30]. The current study cuts across different industries unlike the others [22, 24–26] that focused on a specific industry. We acknowledge the specificity of recommendations that come with studying specific industries but there is also a need for an overarching view of industrial injuries and their associated factors. Also, to the best of the authors' knowledge, no study has estimated occupational injuries among workers from different occupations in the Tema Metropolis, the biggest industrial enclave in Ghana. The lack of evidence may affect the urgency to prioritize promoting employees' health and safety in industrial cities in Ghana. Hence, this study estimated occupational injury prevalence and its determinants among workers in different occupational groups (such as Port and Harbour, Manufacturing Companies, Tourism and Hospitality, Pharmaceuticals etc.). Estimating injury rates for the different occupational groups gives us a snapshot of their burden to inform policy direction towards an appropriate preventive strategy. Also, this will help to improve the health and safety of workers in their working environment in Ghana and beyond. Continually measuring injury prevalence and its associated factors among workers in an industrial city will also help to strengthen the body of evidence that policymakers, health and safety managers and other key stakeholders need to guide occupational health and safety.

## Methods

### Study design and setting

This was a community-based cross-sectional survey among workers in the Tema Metropolis, an industrial setting in Ghana from 22nd December 2020 to 27th February 2021. The adoption of this study design is justified by its ability to estimate both prevalence and inferential statistics like odds ratio which shows the strength of association between dependent (occupational injuries) and independent (such as age, sex, type of engagement etc.) variables [31].

The Tema Metropolis was selected because it has the majority of industries in the Greater Accra Region and also serves as the industrial hub of Ghana. It houses a greater proportion of industrial workers with over 500 industries that produce chemicals, electronics, electrical equipment, furniture, refined oil, aluminium and steel. It has one of the country's two harbours and port services which serves as the convergence point for several industries,

culminating in risks for mechanical and other injury types as well as noise and air pollution. It also has a free zone enclave which is used for the production of goods for both export and local consumption.

## Study population, sample size estimation and sampling

The target population for this study consisted of all workers who were working and resident in the Tema Metropolis of the Greater Accra Region of Ghana. The selection of participants into the study was strictly based on satisfying the inclusion criteria which were being between 18 to 60 years old and having been working in Tema for at least twelve months preceding the study. We excluded those 17 years and below because, in Ghana, the minimum age for engaging a person at work which will predispose him or her to hazard is 18 years [32]. All workers in administrative positions such as receptionists, accounting staff, and secretaries were excluded from the study. These workers were excluded from the study because they were not directly involved in major activity at the workplace, even though, they are also predisposed to musculo-skeletal strain.

Based on the Cochran formula [33], (*sample size* $= (Z^2 \ (P)(1-P))/E^2$) the sample size was determined for the study. Using the prevalence of occupational injuries among workers in Accra Brewery Limited of 36.7% [34], with a 95% confidence interval, approximately 5% margin of error and a design effect of 1.5, an estimated sample size of 428 was calculated. Using a non-response rate of 7%, a total of 472 industrial workers were recruited for the study.

Participants were recruited based on a two-stage sampling procedure with probability proportional to size (S1 Appendix). In low- and middle-income countries where it is difficult to accurately record individual households, this approach was used [35, 36]. In the first stage of the sampling, eight communities (Communities 1, 2, 3, 4, 7, 8, 9, and 10) in Tema were randomly selected from twenty-five communities through balloting. In the second stage, a simple random sampling technique was used to select households in each community using a random code generator. Households that had industrial workers staying there were identified and numbered in each community. In each community, a listing was done to obtain the number of households with industrial workers. In each selected household, all eligible participants who consented were interviewed. This method was repeated in each randomly selected household in all eight communities until the estimated sample size was obtained. Interviews were scheduled with all respondents mostly in the evenings since at that time they would have returned home.

## Data collection

Data collection was done using a structured questionnaire running on a smartphone with the aid of Kobo Collect. The questionnaire was entirely developed by the investigators through a literature review [10, 17, 29, 37–41] and has not been validated. Data was collected on socio-demographic variables including age, sex, educational level, monthly salary, type of engagement, health and safety practices such as usage of PPE, frequency of PPE use, access to health and safety training and occurrence of occupational injuries with emphasis on the types, numbers and mechanisms of the injuries. Data on study participants' satisfaction with existing health and safety measures at the workplace was also collected. Satisfaction with health and safety measures at the workplace was evaluated by asking the study participants their overall satisfaction with these measures with a 'yes' (meaning satisfied) or 'no' (meaning not satisfied) response. The usage of PPE was measured by asking participants whether they wore PPE at the workplace when performing a task with a 'yes' or 'no' response while the frequency of PPE usage was assessed by asking participants how often they wore the PPE when performing a

task at the workplace with 'always' or 'sometimes' response. 'Always' meant that the worker wore PPE anytime he or she performed a task while 'sometimes' meant that the worker only wore the PPE as and when he or she deemed it fit (i.e., not wearing it every time a task was performed at the workplace). Occupational injury was operationalised as all injuries sustained while performing a task at the workplace as defined by the International Labour Organization [42]. To avoid misclassification of occupational injury, study participants who reported sustaining work-related injuries were asked to describe how and where the injury occurred.

The questionnaire was administered in a face-to-face interview in English or the local Twi language by four trained research assistants. The research assistants were all final-year postgraduate Occupational Health and Safety students from the School of Public Health, Kwame Nkrumah University of Science and Technology, Ghana with previous experience in quantitative data collection. They were trained for 3 days by an expert in quantitative data collection and a biostatistician on community entry, consent, the use of Kobo Collect, the selected sampling technique as well as uniform translation of the questions into the local Twi language to ensure consistency in the data collected. The study adopted robust measures such as pre-testing, representative sampling (two-stage sampling), engagement of key stakeholders (epidemiologist, biostatistician, health and safety officers, language expects for both translation and back-translation of the questionnaire into the local Twi language and English respectively) in developing the questionnaire. This was done to improve its reliability and internal validity. The questionnaire was pretested among 50 workers in the North Industrial Area enclave in Accra which is 36 km away from Tema. Based on the feedback from the pre-test, the study tools were modified to suit the objectives of the study before it was used in the main study.

## Data quality assurance

Data quality assurance was conducted by the study biostatistician. The data was downloaded from the Kobo Collect in Excel format and exported to Stata version 16.0 (StataCorp, College Station, USA) for quality management (data cleaning and coding) and analysis. The data was checked for completeness and consistency.

## Statistical analysis

Summary statistics were conducted and presented as means with standard deviations for continuous variables, and frequencies and percentages for categorical variables. The primary outcome was a self-reported occupational injury in the twelve months prior to the interview. Bivariate and multivariate logistic regression analyses were employed to identify the factors that were independently associated with the occurrence of occupational injuries. All covariates that were significantly associated with occupational injury in the univariate analysis and those that were insignificant but considered important or potentially confounding based on previous studies [25, 29, 37, 38, 41, 43] were incorporated in the final multivariate logistic regression model. We controlled for covariates that were insignificant but deemed important in the backward stepwise regression model using a p-value of 0.1. For the bivariate and the final multivariate models, the statistical significance level was kept at a p-value of $< 0.05$ and odds ratios were presented with a 95% confidence level. An interaction term was used to test for the presence of potential effect modification [44].

A multicollinearity test was conducted to confirm whether the explanatory variables that were included in the backward stepwise logistic regression had a correlation using the Variance Inflation Factor (VIF). The results indicated that there was no evidence of multicollinearity between the explanatory variables (mean VIF = 1.63, Maximum VIF = 2.13, Minimum

VIF = 1.12) (See S2 Appendix). Model fit was assessed using the Hosmer-Lemeshow test which showed that the model was good (p = 0.091).

### Ethical consideration

Ethical approval for the study was granted by the Committee on Human Research, Publications and Ethics (CHRPE), School of Medicine and Dentistry, Kwame Nkrumah University of Science and Technology, Kumasi, Ghana (reference number: CHRPE/AP/502/20) after written approval from the Tema Metropolitan Assembly had been obtained before the commencement of the study. Additionally, written informed consent was obtained from all the study participants. Participants' data were completely anonymized using study codes to ensure confidentiality.

## Results

### Socio-demographic characteristics of study participants

Table 1 summarizes the socio-demographic characteristics of the participants. Out of a total of 472 workers that were recruited into the study, 459 completed the questionnaire, representing a response rate of 97.2%. The mean age (±SD) of study workers was 33.9 (±6.8) years and 63.0% of them were married. About 71.0% of the workers were males and 53.6% had tertiary education. Eight out of ten (80.9%) of the workers were permanently engaged at their workplace. About 22.2% of the workers were healthcare workers [Table 1].

### Occupational injuries among the study participants

Table 2 summarizes occupational injuries among the study participants. The proportion of study participants who experienced an occupational injury in the last 12 months was 64.7% (95%CI: 0.60–0.69). Out of the 297 workers that were injured, over two-thirds (64.3%) suffered only one occupational injury. Approximately 74.0% of them reported the occurrence of accidents. About 34.0% of the injuries were cuts/punctures. About 45.8% of the workers indicated that they were injured by the work tool [Table 2].

### Occupational health and safety practices

Table 3 summarizes the occupational health and safety practices among the study participants. Approximately 46.0% of the workers indicated they had a health and safety department at their workplace. More than half (51.6%) of them had received training on health and safety practices. Over 22.7% (104) of the workers indicated that they used PPE at the workplace. Out of the 104 workers who indicated they used PPE, about 24.0% indicated they used it always at the workplace. Approximately 15.2% of the workers were satisfied with health and safety measures at the workplace [Table 3].

### Factors associated with occupational injuries among study participants

Table 4 summarizes the factors associated with occupational injuries among the study participants. In the univariate analysis, factors such as age group, sex, monthly salary, type of engagement, educational level, industry type, use of PPE, industry type, health and safety training, and satisfaction with health and safety were associated with occupational injuries.

After adjusting for significant variables in the backward stepwise logistic regression, workers who were casual staff (AOR: 2.26, 95%CI: 1.04–4.92), working at port and harbour (AOR: 3.77, 95%CI: 1.70–8.39), monthly salary of 1000 to 1500 cedis (188.68–257.29 USD) (AOR: 2.59, 95%CI: 1.40–4.77), no health and safety training (AOR: 2.18, 95%CI: 1.08–4.39) and

**Table 1. Socio-demographic characteristics of study participants.**

| Variables | Frequency, N = 459 | Percentage, % [Range] |
|---|---|---|
| **Age group (years)** | | |
| 20–29 | 140 | 30.5 |
| 30–39 | 211 | 46.0 |
| 40+ | 108 | 23.5 |
| **Mean age (±SD)** | 33.9 (±6.8) | [21 – 53] |
| **Sex** | | |
| Male | 326 | 71.0 |
| Female | 133 | 29.0 |
| **Marital status** | | |
| Married | 289 | 63.0 |
| Single | 159 | 34.6 |
| Divorced | 11 | 2.4 |
| **Educational Level** | | |
| Basic | 84 | 18.3 |
| Secondary | 129 | 28.1 |
| Tertiary | 246 | 53.6 |
| **Religion** | | |
| Christian | 338 | 73.6 |
| Muslim | 121 | 26.4 |
| **Type of engagement** | | |
| Permanent | 369 | 80.4 |
| Casual | 90 | 19.6 |
| *Monthly salary | | |
| < 1000 | 92 | 20.0 |
| 1000–1500 | 200 | 43.6 |
| 1501–2000 | 116 | 25.3 |
| > 2000 | 51 | 11.1 |
| **Industry type** | | |
| Agriculture and Forestry | 44 | 9.6 |
| Construction | 43 | 9.4 |
| Manufacturing | 55 | 12.0 |
| Port & Harbour | 83 | 18.1 |
| Tourism & Hospitality | 19 | 4.1 |
| Communication & Commerce | 29 | 6.3 |
| Health Care | 102 | 22.2 |
| Pharmaceuticals | 46 | 10.0 |
| Others[b] | 38 | 8.3 |
| **Hours spent at the workplace (n = 215)** | | |
| 4–6 | 2 | 0.9 |
| 7–9 | 90 | 41.9 |
| 10–12 | 123 | 57.2 |

SD: Standard deviation

[a]Analyzed using Fisher's exact test

[b]Include Transport, Waste Management Services, Janitorial, Textiles

*GHC 5.83: USD 1.00 per exchange rate in Ghana during the study period

**Table 2. Occupational injuries among study participants.**

| Variables | Frequency, N = 459 | Percentage, % |
|---|---|---|
| **Occupational injury in the last 12 months** | | |
| Yes | 297 | 64.7 |
| No | 162 | 35.3 |
| **Number of times injured (n = 297)** | | |
| 1 | 191 | 64.3 |
| 2 | 91 | 30.6 |
| 3+ | 15 | 5.1 |
| **Types of injury (n = 297)** | | |
| Cut/puncture | 101 | 34.0 |
| Abrasion | 90 | 30.3 |
| Burns | 41 | 13.8 |
| Dislocation | 31 | 10.4 |
| Fracture | 30 | 10.1 |
| Crashing | 3 | 1.0 |
| Eye injury | 1 | 0.3 |
| **Mechanism of injury (n = 297)** | | |
| Working tool | 136 | 45.8 |
| Hot surface, substance or chemical | 42 | 14.1 |
| Collision | 40 | 13.5 |
| Falling object (s) | 29 | 9.8 |
| Falls | 24 | 8.1 |
| Moving object (s) | 17 | 5.7 |
| Lifting heavy object | 9 | 3.0 |
| **Reported accidents (n = 297)** | | |
| Yes | 219 | 74.0 |
| No | 77 | 26.0 |

dissatisfaction with health and safety measures (AOR: 4.31, 95%CI: 2.12–8.78) were associated with the occurrence of occupational injuries. Having tertiary education (AOR: 0.03, 95%CI: 0.01–0.10) and secondary education (AOR: 0.22, 95%CI: 0.07–0.74) were protective of occupational injuries compared to basic education [Table 4].

## Discussion

The economic growth and development of every nation are largely dependent on the working population. It is therefore important to create an atmosphere at the workplace that enhances the health and safety of the employees given their significant contribution to achieving organizational goals. This study sought to estimate the prevalence of occupational injuries, identify their determinants, and to also assess occupational health and safety practices among workers. The study found a high prevalence of occupational injuries. In terms of the factors associated with occupational injuries, the results in this study collaborate with the constrain-response accident model which uses both proximal and distal factors to explain the occurrence of occupational injuries [18]. Hence, we adopted this model to explain the proximal (including educational level, industry type, and satisfaction with health and safety measures) and distal (such as health and safety training, monthly income and type of engagement) factors predicting occupational injuries in this study.

**Table 3. Occupational health and safety practices among the study participants.**

| Variables | Frequency, N = 459 | Percentage, % |
|---|---|---|
| **Health and Safety department** | | |
| Yes | 211 | 46.0 |
| No | 182 | 39.7 |
| Not sure | 66 | 14.4 |
| **Health and Safety training** | | |
| Yes | 237 | 51.6 |
| No | 222 | 48.4 |
| **Frequency of training (n = 237)** | | |
| Annually | 98 | 41.4 |
| Biannually | 82 | 34.6 |
| Quarterly | 31 | 13.1 |
| Alternate Months | 16 | 6.8 |
| No Definite Time | 10 | 4.2 |
| **PPE usage at work** | | |
| No | 355 | 77.3 |
| Yes | 104 | 22.7 |
| **Frequency of PPE use at work (n = 104)** | | |
| Sometimes | 79 | 76.0 |
| Always | 25 | 24.0 |
| **Satisfied with health and safety measures at the workplace*** | | |
| No | 387 | 84.3 |
| Yes | 72 | 15.7 |

NB: Except for 'frequency of training' and 'frequency of PPE use', N was 459 for all variables

*Self-reported

The prevalence of occupational injuries observed in the present study was high. This is higher than the 57.9% and 47.9% prevalence rates that were previously reported among construction workers and welders in Ghana respectively [16, 45]. It is worth noting that there is no study in Ghana which estimated occupational injuries among workers in different occupational groups which makes it difficult to directly compare the prevalence of occupational injury in the present study. Despite that, the prevalence of occupational injuries observed among workers in the present study provides enough evidence of a high burden of occupational injuries among workers from different occupational groups in an industrial city in Ghana. Also, the findings of this study give a bird's eye view of the subject matter in Ghana. The differences in the prevalence of occupational injury could be explained by the difference in study settings, populations (workers from different occupational groups), working conditions and the types of hazards exposed to at the workplace. Hence, the present study provides a more reliable and functional overview of the burden of the phenomenon among different occupational groups compared to the earlier studies which focused on only one occupation [16, 45].

The present study is consistent with a study in Ethiopia that reported about 63.4% occupational injury prevalence among industrial workers [46]. However, in an urban city in India, Sashidharan and Gopalakrishnan (2017) in a study among industrial workers reported an injury prevalence of 44.3% [37] which is lower than the prevalence rate (63.4%) observed in our study. In the present study, we estimated injury prevalence among over nine different occupational groups compared to two in the study in India [37] which could affect the injury

**Table 4. Factors associated with occupational injuries among the study participants.**

| Variables | Injuries | Bivariate | | Multivariate | |
|---|---|---|---|---|---|
| | n (%) | OR (95%CI) | P–value | AOR (95%CI) | P–value |
| **Age group (years)** | | | | | |
| 20–29 *(Ref)* | 100 (71.4) | 1.00 | | - | - |
| 30–39 | 122 (57.8) | 0.55 (0.35–0.87) | 0.010 | - | - |
| 40+ | 75 (69.4) | 0.91 (0.52–1.58) | 0.734 | - | - |
| **Sex** | | | | | |
| Female *(Ref)* | 63 (47.4) | 1.00 | | - | - |
| Male | 234 (71.8) | 2.83 (1.86–4.29) | <0.001 | - | - |
| **Educational Level** | | | | | |
| Basic *(Ref)* | 79 (94.1) | 1.00 | | 1.00 | |
| Secondary | 106 (82.2) | 0.29 (0.11–0.80) | 0.017 | 0.22 (0.07–0.74) | 0.015 |
| Tertiary | 112 (45.5) | 0.05 (0.21–0.14) | <0.001 | 0.03 (0.01–0.10) | <0.001 |
| **Type of engagement** | | | | | |
| Permanent *(Ref)* | 223 (60.4) | 1.00 | | 1.00 | |
| Casual | 74 (82.2) | 3.03 (1.70–5.40) | <0.001 | 2.26 (1.04–4.92) | 0.039 |
| **Monthly salary (cedis)** | | | | | |
| < 1000 *(Ref)* | 74 (80.4) | 1.00 | | 1.00 | |
| 1000–1500 | 141 (70.5) | 0.58 (0.32–1.06) | 0.075 | 2.59 (1.40–4.77) | 0.002 |
| 1501–2000 | 59 (50.9) | 0.25 (0.13–0.47) | <0.001 | - | - |
| > 2000 | 23 (45.1) | 0.20 (0.09–0.42) | <0.001 | - | - |
| **Industry type** | | | | | |
| Agriculture and Forestry *(Ref)* | 32 (72.7) | 1.00 | | 1.00 | |
| Construction | 35 (81.4) | 1.64 (0.59–4.53) | 0.339 | - | - |
| Manufacturing | 43 (78.2) | 1.34 (0.53–3.38) | 0.530 | - | - |
| Port & Harbour | 64 (77.1) | 1.26 (0.55–2.92) | 0.585 | 3.77 (1.70–8.39) | 0.001 |
| Tourism & Hospitality | 5 (26.3) | 0.13 (0.04–0.45) | 0.001 | 0.09 (0.02–0.33) | <0.001 |
| Communication & Commerce | 6 (20.7) | 0.10 (0.03–0.30) | <0.001 | 0.03 (0.01–0.11) | <0.001 |
| Health Care | 67 (65.7) | 0.72 (0.33–1.56) | 0.404 | - | - |
| Pharmaceuticals | 33 (71.7) | 0.95 (0.38–2.40) | 0.917 | 2.48 (1.03–5.99) | 0.043 |
| Others | 12 (31.6) | 0.17 (0.07–0.45) | <0.001 | 0.10 (0.03–0.36) | <0.001 |
| **Use of PPE at work** | | | | | |
| Yes *(Ref)* | 44 (42.3) | 1.00 | | 1.00 | - |
| No | 253 (71.3) | 3.38 (2.15–5.31) | <0.001 | 2.62 (0.97–7.02) | 0.056 |
| **Health and safety training** | | | | | |
| Yes *(Ref)* | 148 (62.5) | 1.00 | | 1.00 | |
| No | 149 (67.1) | 1.23 (0.84–1.80) | 0.296 | 2.18 (1.08–4.39) | 0.029 |
| **Satisfied with health and safety measures at the workplace** | | | | | |
| Yes | 28 (38.9) | 1.00 | | 1.00 | |
| No | 269 (69.5) | 3.58 (2.13–6.03) | <0.001 | 4.31 (2.12–8.78) | <0.001 |

OR: Odds ratio

AOR: Adjusted Odds ratio

Ref: reference category

rates. Despite that, all these studies show a high prevalence of occupational injuries among workers. Hence, there is a need for policy direction such as regular maintenance of work tools and frequent use of PPE at the workplace to reduce the burden of occupational injuries among workers. Among workers in this study who experienced an occupational injury, the commonly

reported types were cuts/punctures, abrasions, burns, fractures and dislocation, similar to what was reported in an Ethiopian study [41]. It was also observed in the present study that the major mechanism of injury was the use of working tools (45.8%). Hot surfaces, substances or chemicals, collisions and falling objects also contributed to the occurrence of occupational injuries among study participants. These were similar to previous studies that reported working tools, injury from falling objects, and collision as mechanisms of occupational injuries [30, 39, 40, 47]. This underscores the importance of repeated training of workers on the safe use and storage of tools with high injury-causing potential.

The proximal factors (individual and socio-demographic factors) which had a significant effect on the occurrence of occupational injuries included educational level, industry type, and satisfaction with health and safety measures. The present study observed that a higher educational level was associated with reduced odds of occupational injury which resonates with studies conducted in Ghana, Ethiopia, Iran and Canada [30, 43, 48, 49]. The workers with tertiary and secondary education had 97.0% and 78.0% reduced odds of experiencing an occupational injury respectively compared to those that had basic education. An attributable reason may be that by attaining higher education, an individual may have a better appreciation, understanding and observation of health and safety practices, thereby reducing his or her risk of occupational injury. Education could also be capturing some level of language barrier especially when most of these safety signs and guidelines are written in English which may cause a challenge in communication. Secondly, workers with tertiary education are more likely to be managerial-level staff and thus less engaged in high-risk jobs or activities at the workplace.

Additionally, the type of occupation was found to be a significant determinant of occupational injury at the workplace. It was observed in the present study that workers who worked at the Port and Harbour had about 4 times (AOR: 3.77, 95%CI: 1.70–8.39) increased odds of experiencing occupational injury compared to those at Agriculture and Forestry. Variations in levels of exposure to occupational hazards in the two settings may account for this. The nature of activities at the Port and Harbour may expose them to occupational hazards such as the lifting of heavy objects and working close to dangerous equipment.

The majority (84.3%) of the workers in the present study were dissatisfied with management's commitment to their health and safety and those who were dissatisfied had about four times (AOR: 4.31, 95%CI: 2.12–8.78) increased odds of occupational injury. This is particularly worrying because the management of every occupation plays a significant role in the prevention of occupational injury. This can be achieved by setting up policies that will enhance employees' health and safety [50]. Employees who are satisfied with the health and safety policies or culture at the workplace are more likely to adhere to regulations, an essential step in preventing accidents and injuries [51, 52].

The distal factors are the organizational and work-related characteristics which had a significant influence on the occurrence of occupational injuries among the workers. Specifically, this study found that health and safety training, monthly income and type of engagement were significantly associated with the occurrence of occupational injuries among the workers. The risk of occupational injury was higher in casual workers compared to permanent workers. We observed in the present study that casual workers had about two times (AOR: 2.26, 95%CI: 1.04–4.92) increased odds of occupational injury compared to permanent workers. This is in agreement with earlier reports that casual workers are susceptible to an increased risk of occupational injuries compared to permanent workers [53, 54]. In Ethiopia, a study among industrial workers reported that casual workers were about 7 times more likely to experience occupational injury compared to permanent workers [46]. Casual workers are mostly inexperienced, less knowledgeable about occupational hazards and usually do not benefit from health and safety training at the workplace, predisposing them to occupational injuries [55, 56].

Health and safety training is one of the important strategies for the prevention of occupational injuries among workers [20]. It was also observed in the present study that, workers who did not receive training in health and safety had about two times (AOR: 2.18, 95%CI: 1.08–4.39) increased odds of occupational injury compared to their counterparts who received training in health and safety. This is supported by the findings of a study in Ethiopia which reported that compared to workers who received training in health and safety, those who did not had about four times increased risk of occupational injuries [38]. A possible reason could be that receiving training on health and safety will enhance the knowledge and awareness of the workers about the possible exposures to workplace hazards and teach them how to protect themselves. Also, the training on health and safety may help to shape their attitudes and behaviours towards health and safety at the workplace. We, therefore, advocate for training and re-training of workers on health and safety at the workplace looking at its significant effect on injury prevention at the workplace.

It was also observed in this study that monthly salary was significantly associated with occupational injury. Workers who were on a monthly salary of 1000 to 1500 cedis (188.68–257.29 USD) had about three times increased odds (AOR: 2.59, 95%CI: 1.40–4.77) of occupational injuries compared to their counterparts on <1000 cedis (<188.68 USD). This finding resonates with an earlier study in Ghana where workers on high income (233.80–350.50 USD) had about four times higher odds of occupational injuries compared to those on low income (84.20–116.70 USD) [16]. However, this finding is at variance with a study in Ethiopia where workers with low salaries had an increased risk of occupational injury compared to those with high salaries [57]. The current study reports on occupational injuries among workers in different institutions with different salary structures and occupational exposures compared to the Ethiopian study which reported injury prevalence among workers in the same occupation.

## Strengths and limitations of the study

Though other studies in Ghana [16, 27–30] have estimated injury prevalence among workers in specific populations, this is the first study to estimate the prevalence of occupational injuries which involve a wide range of workers from different industries. Hence, the findings are quite representative of the different occupational groups. Another, strength of the present study is the reduction of the healthy-worker bias due to the adoption of the household approach in sampling the study participants.

The study had some limitations. The approach of participant self-reporting of occupational injuries and the use of PPE could have affected the reported prevalence of occupational injuries and PPE use. Again, the use of self-report approach could also lead to the introduction of social desirability bias. There could have been an element of recall bias since study participants had to do a retrospective (past 12 months) report of occupational injury. However, study participants were asked to give an account of how the injury occurred to minimize the introduction of recall bias. Again, this study did not report on how much time was lost from work due to the injury sustained at the workplace. Variables such as "monthly salary of 1000–1500 cedis", "ports and harbour", "pharmaceuticals" and "health and safety training" had their adjusted odds ratio in the final multivariate logistic regression model increased from their crude odds ratio estimates. This could have been affected by some missing data in the other variables that were considered for the final logistic regression model or a confounder. The results in this study, however, are comparable to earlier studies and offer useful information about the factors contributing to occupational injuries.

Despite Tema being the largest industrial hub in Ghana, data obtained from only one site limits the generalizability of the study findings. However, since the data were collected from

eight communities in Tema, the study findings can be generalized to this population. Another limitation of this study was the inability to estimate the long-term injury risk (for instance, adverse health outcomes in the future from present-day inhalation of gases) of the study participants. The data collection instrument was not intentionally validated. However, pretesting, the adoption of a standard definition of occupational injury, and a simple random sampling approach could reduce biases.

## Conclusion

There is a high prevalence of occupational injuries among the study participants and these were mainly caused by working tools and hot surfaces, substances or chemicals. Factors such as employees' level of education, type of engagement, monthly income, type of industrial work, health and safety training and satisfaction with health and safety measures at the workplace were independently associated with occupational injuries. The Ministry of Trade and Industry, Department of Factories Inspectorate, and National Labour Commission of Ghana should collaborate and conduct routine monitoring and evaluations at various institutions (especially those in industrial cities) to ensure that employees' health and safety are protected at the workplace. We also recommend that the Association of Ghana Industries (AGI) should create more awareness of employees' health and safety by admonishing their members to be conscious of the need to adhere to safety practices at the workplace. Future studies can adopt a mixed-method approach to explore the satisfaction of health and safety policies at the workplace and how these contribute to occupational injuries.

## Supporting information

**S1 Appendix. Sampling approach of study participants.**
(DOCX)

**S2 Appendix. Table: Multi-collinearity test results.**
(DOCX)

**S1 Dataset.**
(XLS)

## Author Contributions

**Conceptualization:** Michael Tetteh Asiedu, Douglas Aninng Opoku, Nana Kwame Ayisi-Boateng, Emmanuel Kweku Nakua.

**Data curation:** Michael Tetteh Asiedu, Douglas Aninng Opoku, Nana Kwame Ayisi-Boateng, Joseph Osarfo, Alhassan Sulemana, Jennifer Ashilevi, Felix Agyemang Opoku.

**Formal analysis:** Douglas Aninng Opoku, Joseph Osarfo, Aliyu Mohammed.

**Methodology:** Michael Tetteh Asiedu, Douglas Aninng Opoku, Nana Kwame Ayisi-Boateng, Joseph Osarfo, Alhassan Sulemana, John Amissah, Jennifer Ashilevi, Ayongo Mate-Kole, Felix Agyemang Opoku, Emmanuel Kweku Nakua.

**Project administration:** Michael Tetteh Asiedu, Douglas Aninng Opoku, Nana Kwame Ayisi-Boateng, Joseph Osarfo, Alhassan Sulemana, John Amissah, Ayongo Mate-Kole, Isaac Kofi Yankson.

**Supervision:** Emmanuel Kweku Nakua.

**Validation:** Michael Tetteh Asiedu, Douglas Aninng Opoku, Joseph Osarfo, Alhassan Sulemana, Aliyu Mohammed.

**Writing – original draft:** Michael Tetteh Asiedu, Douglas Aninng Opoku, Nana Kwame Ayisi-Boateng, Joseph Osarfo, Aliyu Mohammed, Isaac Kofi Yankson, Emmanuel Kweku Nakua.

**Writing – review & editing:** Michael Tetteh Asiedu, Douglas Aninng Opoku, Nana Kwame Ayisi-Boateng, Joseph Osarfo, Alhassan Sulemana, Aliyu Mohammed, John Amissah, Jennifer Ashilevi, Ayongo Mate-Kole, Felix Agyemang Opoku, Isaac Kofi Yankson, Emmanuel Kweku Nakua.

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
