## [Decision Letter · Decision Letter 0]

9 May 2023

PONE-D-23-10198Epidemiology of occupational injuries in an industrial city in Ghana: A cross-sectional surveyPLOS ONE

Dear Dr. Opoku,

Thank you for submitting your manuscript to PLOS ONE. After careful consideration, we feel that it has merit but does not fully meet PLOS ONE’s publication criteria as it currently stands. Therefore, we invite you to submit a revised version of the manuscript that addresses the points raised during the review process.

We look forward to receiving your revised manuscript.

Kind regards,

Aiggan Tamene

Academic Editor

PLOS ONE

Journal Requirements:

Additional Editor Comments (if provided):

Title: Major Revision Required - "Epidemiology of Occupational Injuries in an Industrial City in Ghana: A Cross-Sectional Survey"

Dear Authors,

Thank you for submitting your manuscript titled "Epidemiology of Occupational Injuries in an Industrial City in Ghana: A Cross-Sectional Survey" for consideration. After carefully reviewing your work, I believe there are several major revisions required to improve the clarity, rigor, and overall quality of your study. I have outlined these revisions below:

1. Methodology and Study Design:

• Provide a more detailed description of the study design, including the sampling method employed and the rationale behind it. This will enhance the transparency and reproducibility of your study.

• Clearly define the inclusion and exclusion criteria for selecting participants. This will help readers understand the population under investigation and improve the study's generalizability.

• Elaborate on the data collection process, including the tools used, the training provided to data collectors, and the steps taken to ensure data quality and minimize bias.

2. Statistical Analysis:

• Describe the statistical methods employed in analyzing the data more comprehensively. Specify the statistical tests used, including appropriate adjustments for confounding variables and potential sources of bias.

• Include details on the software used for statistical analysis, along with the version number, to facilitate reproducibility.

3. Results and Discussion:

• Present the results in a clear and organized manner, using appropriate tables and figures to enhance the readability and interpretation of the findings.

• Discuss the limitations of the study, including any potential biases or confounding factors that may have influenced the results. This will provide a more comprehensive understanding of the implications of your findings.

• Compare your results with existing literature on occupational injuries in similar settings to contextualize your findings and highlight the novel contributions of your study.

4. Conclusion:

• Revise and strengthen the concluding statements to accurately reflect the key findings of your study.

• Clearly state the implications of your research and highlight potential avenues for future research and intervention strategies.

5. Language and Clarity:

• Carefully proofread the manuscript to address grammatical errors, awkward phrasing, and typographical mistakes.

• Ensure consistency in terminology and abbreviations throughout the manuscript.

Please address these major revisions in your manuscript. I believe that incorporating these changes will significantly improve the scientific rigor and impact of your study. Once the revisions have been made, I would be delighted to reevaluate your manuscript for further consideration.

Thank you for your attention to these matters. I look forward to receiving your revised manuscript.

Reviewers' comments:

Reviewer's Responses to Questions

**Comments to the Author**

1. Is the manuscript technically sound, and do the data support the conclusions?

Reviewer #1: No

Reviewer #2: Yes

Reviewer #3: Partly

2. Has the statistical analysis been performed appropriately and rigorously? 

Reviewer #1: No

Reviewer #2: Yes

Reviewer #3: Yes

3. Have the authors made all data underlying the findings in their manuscript fully available?

Reviewer #1: Yes

Reviewer #2: Yes

Reviewer #3: Yes

4. Is the manuscript presented in an intelligible fashion and written in standard English?

Reviewer #1: Yes

Reviewer #2: Yes

Reviewer #3: Yes

5. Review Comments to the Author

Reviewer #1: I appreciate the invitation to review the manuscript. The manuscript was reviewed and general comments about the subject and content of paper was presented as follow. Based on our review, it seems that the mentioned paper is not suitable for publication in PLOS ONE.

• The content of paper is not enough for publication in PLOS ONE

• It is not observed new finding in this research.

Reviewer #2: Reviewer’s comments

Introduction

Page 3, line 61

“The global burden of occupational injuries is estimated to be 1.9 million deaths in 2019” is this data/figure includes occupational diseases OR occupational injuries only?

Pages 4, line 88-92

The statement is about the associated factors; as result it is better to merge with the previous paragraph.

Pages 4, line 93

“Apart from the few studies mentioned above,…”

What your study added to this few studies?

Methods

Pages 5, line 123

Why you use 1.2 as design effect? Why not 1.5 or 2?

Could you support with credible source/evidence

Pages 6, line 150-152

“’Occupational injury’ was operationalized as all injuries sustained while performing a task at the workplace that resulted in a loss of at least 12 hours of productive time”

How did you confirm the impact of injury (a loss of at least 12 hours of productive time)?

Pages 6, line 160-161

Are you used Chi square or Fisher’s exact tests analysis? Better to state what you already used

Results

Pages 7, line 185-186

“The proportion of study participants that experienced an occupational injury in the last 12 months

was 64.7%.”

State the proportion of occupational injuries with 95% CI; the lower and upper confidence intervals

Pages 8, line 197-199

“Over 54.9% of them were supplied with PPE and 77.3% of the study participants indicated that they used PPE at the workplace”

How the proportion of PPE use is much higher than the proportion of PPE supply (77.5% vs 54.9%)

Table 2

Pages 9, line 207-209

For the variable “use of PPE at work”; you reported as an about 355 (77.3) are not using PPE. However in the variable “Frequency of PPE use at work” you characterize the frequency of use of PPE for 355 study participants. You are expected to characterize the 104 study participants who reported as they use PPE

How this can be? It is not clear are not inline. Please re consider the presentation of “use of PPE at work and Frequency of PPE use at work”

Pages 10, line 230-232

“Workers that were working at the Port and Harbour were about 4 times (AOR: 4.32; 95%CI: 0.14 – 16.37) more likely to experience occupational injury compared to those at Agriculture and Forestry”

The CI contains the null value (1), therefore, how this variable was considered as statistically significant covariate?

Pages 11&12, line 239-241

Have you checked the assumptions of Binary logistic regression before conducting analysis?

The variable” Frequency of PPE use at work” is not collected from all study participant rather only from 104 study participants; who reported as they used PPE. As result, can it can be regressed against Occupational injury?

In the logistic regression analysis table (table 4), the outcome variable (occupational injury) with the number of injured and non-injured study participants shall be presented in a column, next to the list of variables column.

Discussion

Pages 13, line 291-293

“It is recommended that there should be reduced payment benefits to the affected worker if investigations reveal he or she was not in PPE at the time of injury.”

It shall be supported with credible evidence. What if the organization didn’t supply adequate PPE and workers unable to purchase and use by their own?

Overall comment

Please work to improve the quality of the writing throughout the manuscript

Reviewer #3: The manuscript, in its present form, contains several weaknesses. Appropriate revisions to the following points should be undertaken to justify the recommendation for publication.

Abstract

Page 2 line 27, please paraphrase it.

Page 2 lines 43-44: try to reanalyze and interpret the association of your outcome variable with the level of education.

Page 2 line 45: “Working at Port 45 and Harbour (AOR: 4.32; 95%CI: 0.14 – 16.37)” Does it show a significant association?

Introduction

The magnitude of the problem should be clearly stated in an inverted-triangular form (global to local perspective).

The authors should clearly show the research gap that they want to fill with the present study. They should intensively review previous research conducted in the area and direct how they will generate new evidence with their study.

Method

The study setting and area should be mentioned precisely, even with the relative and absolute location. Therefore, a map of the study area should be included.

Page 5 lines 118-119: “All workers who reported non-occupational related 119 injuries and …… were excluded.” Do you mean all included subjects were those who have an occupational injury (Prevalence =100%)???

Page 5, line 123: why do you use a design effect of 1.2, which is nearly 1.00? What is the need to use the design effect? You have used two-stage sampling. What do you think about the proper design effect to be used for two-stage sampling? Please try to address it.

If there is more than one industrial worker in a household, how do you select the respondents? What inclusion and exclusion criteria were used during sampling?

How to control the reported bias (in case, some respondents may respond as if they were injured at the workplace, while it actually occurred in non-occupational areas)?

Why do you use the KOBO toolbox and ODK data collection tools in combination? Use of the one will be simple for the management of the data and training of data collectors.

Some of your variables, for instance, the use of PPE, should be assessed through observation, not using a questionnaire. It raises the validity question of your finding.

Result:

Page 10, line 231: Please check your presentation regarding the association of level of education with occupational injuries.

Page 11: On tabular presentation of the logistic regression,

- You should include the frequency and percentage of study participants per the dichotomous outcome variable (injured/non-injured)

- The P-value for the crude odds ratio is not important.

Discussion

Page 12, line 249: the prevalence is much higher than 50%, therefore, the phrase “more than half” is not descriptive of your finding.

The discussion and conclusions are well written.

6. PLOS authors have the option to publish the peer review history of their article (what does this mean?). If published, this will include your full peer review and any attached files.

Reviewer #1: No

Reviewer #2: **Yes: **Giziew Abere

Reviewer #3: No

---

## [Author Response · Author response to Decision Letter 0]

16 Jun 2023

Academic Editor 

Comment: Provide a more detailed description of the study design, including the sampling method employed and the rationale behind it. This will enhance the transparency and reproducibility of your study.

Response: This has been done as suggested. 

This was a community-based cross-sectional survey among workers in the Tema Metropolis, an industrial setting in Ghana from 22nd December 2020 to 27th February 2021. The adoption of this study design is justified by its ability to estimate both prevalence and inferential statistics like odds ratio which shows the strength of association between dependent (occupational injuries) and independent (age, sex, type of engagement etc.) variables (28). [Page 4, Lines 110-113]

Comment: Clearly define the inclusion and exclusion criteria for selecting participants. This will help readers understand the population under investigation and improve the study's generalizability

Response: This was present in the initial submission but has been revised to improve clarity. The revised section now reads;

The selection of participants into the study was strictly based on satisfying the inclusion criteria which was being between 20 to 60 years old and having been working in Tema for at least twelve months. All workers in administrative positions (such as receptionists, accounting staff and secretaries) were excluded from the study. [Page 5, Lines 124-127]

Comment: Elaborate on the data collection process, including the tools used, the training provided to data collectors, and the steps taken to ensure data quality and minimize bias.

Response: These were provided initially but have now been made more explicit and clearer.

Data collection was done using a structured questionnaire running on a smartphone with the aid of Kobo Collect. The questionnaire was self-developed entirely new by the investigators through a literature review [10,21,23,33–37]. Data was collected on socio-demographic variables including age, sex, educational level, monthly salary, type of engagement, health and safety practices such as usage of PPE, frequency of PPE use and access to health and safety training and occurrence of occupational injuries with emphasis on the types, numbers and mechanisms of the injuries. The usage of PPE was measured by asking participants whether they wore PPE at the workplace when performing a task with a ‘yes’ or ‘no’ response while the frequency of PPE usage was assessed by asking participants how often they wear the PPE when performing a task at the workplace with ‘always’ or ‘sometimes’ response. ‘Always’ means that the worker wears PPE anytime he or she is performing a task while ‘sometimes’ means that the worker only wears the PPE as and when he or she deemed it fit (i.e. not wearing it all times when performing a task at the workplace. Occupational injury was operationalized as all injuries sustained while performing a task at the workplace as defined by the International Labour Organization [38]. To avoid misclassification of occupational injury, study participants who reported sustaining work-related injuries were asked to describe how and where the injury occurred. 

The questionnaires were administered in a face-to-face interview in English or the local Twi language by four trained research assistants. The research assistants were all final-year post-graduate Occupational Health and Safety students from the School of Public Health, Kwame Nkrumah University of Science and Technology, Ghana with previous experience in quantitative data collection. They were trained for 3 days by an expert in quantitative data collection and a biostatistician on community entry, consent, the use of Kobo Collect, the selected sampling technique as well as uniform translation of the questions into the local Twi language to ensure consistency in the data collected. The questionnaire was pretested among 50 workers in the North Industrial Area enclave in Accra which is 36 km away from Tema. Based on the feedback from the pre-test, the study tools were modified to suit the objectives of the study before it was used in the main study. [Page 6, Lines 147-173]

Comment: Describe the statistical methods employed in analyzing the data more comprehensively. Specify the statistical tests used, including appropriate adjustments for confounding variables and potential sources of bias.

Include details on the software used for statistical analysis, along with the version number, to facilitate reproducibility.

Response: We performed the analysis again using a logistic regression analysis with a backward stepwise approach. This has been provided in the revised manuscript. The specific section now reads as;

Data was cleaned by checking for wrong entries and exported to Stata version 16.0 (StataCorp, College Station, USA) for analysis. Summary statistics were conducted and presented as means with standard deviations for continuous variables, and frequencies and percentages for categorical variables. Associations between predictor (such as age, sex, health and safety training etc.) and outcome (occupational injury) variables were explored using either Chi-square or Fisher’s exact (for proportions with subgroups <5) tests. The primary outcome was a self-reported occupation-related injury in the twelve months prior to the interview. Bivariate and multivariate logistic regression analyses were employed to quantify the strength of association between the occurrence of occupational injuries and their determinants. The effects of other covariates on the outcome were adjusted for in the final model using a backward stepwise approach with a p-value of 0.1. In the multivariate logistic regression model, all predictor variables that had both statistically significant and insignificant association with the outcome variables were taken into account. They were considered important or potentially confounding based on previous studies (20,23,33,34,37,39) in arriving at the final backward stepwise regression model. For the bivariate and the final multivariate models, statistical significance level was kept at a p-value of <0.05 and odds ratios presented at a 95% confidence level. [Page 7, Lines 174-190]

Comment: Present the results in a clear and organized manner, using appropriate tables and figures to enhance the readability and interpretation of the findings.

Response: This has been done as suggested. 

Comment: Discuss the limitations of the study, including any potential biases or confounding factors that may have influenced the results. This will provide a more comprehensive understanding of the implications of your findings.

Response: We controlled for confounding variables in the analysis by adjusting for these variables in the logistic regression analysis. This has been acknowledged in the revised manuscript under subheading ‘’Data management and statistical analysis’’. Again, to reduce information bias, participants were asked to describe how and where the injury occurred and this has been acknowledged in the revised manuscript under subheading ‘’Data collection’’ [Page 6]

Other limitations have been provided under the subheading ‘’strengths and limitations of the study’’ [Page 15, Lines 342-354]

Comment: Compare your results with existing literature on occupational injuries in similar settings to contextualize your findings and highlight the novel contributions of your study.

Response: We compared our findings to a study in Ethiopia which was also conducted among workers in an industrial city in the initial manuscript. In the revised manuscript, we have also compared the findings in the present study to similar studies conducted among workers in industrial city in India. It reads;

However, in an urban city in India, Sashidharan et al (2017) in a study among industrial workers reported an injury prevalence of 44.3% (33) which is lower than the prevalence rate (63.4%) observed in our study. In the present study, we estimated injury prevalence among nine different occupational groups while the study in India (33) estimated injury prevalence among two different occupational groups which could affect the injury rates in the two studies. Despite that, all these studies show a high prevalence of occupational injuries among workers. [Page 13, Lines 271-277]

Comment: Revise and strengthen the concluding statements to accurately reflect the key findings of your study.

Response: This has been done as suggested. The revised section now reads;

There is a high prevalence (64.7%) of occupational injuries among the study participants. Factors associated with this are employees’ level of education, type of engagement, type of industrial work, health and safety training and their level of satisfaction with health and safety measures at the workplace. It is imperative for effective mitigation measures to be instituted for this high-risk population. These include training and retraining in health and safety, education and awareness campaigns, as well as policy enforcement measures among workers, especially, those with a low level of education and temporary staff. [Page 13, lines 356-366]

Comment: Clearly state the implications of your research and highlight potential avenues for future research and intervention strategies

Response: This has been done as suggested. It reads;

The present study provides useful information that can guide efforts in the promotion of health and safety among workers at the workplace. Future studies can adopt a qualitative approach to explore the satisfaction of health and safety policies at the workplace. [Page 13, Lines 363-366]

Comment: Carefully proofread the manuscript to address grammatical errors, awkward phrasing, and typographical mistakes.

Response: Please, this has been done as suggested 

Comment: Ensure consistency in terminology and abbreviations throughout the manuscript

Response: Please, this has been done as suggested 

Reviewer 1

Comment: I appreciate the invitation to review the manuscript. The manuscript was reviewed and general comments about the subject and content of paper was presented as follow. Based on our review, it seems that the mentioned paper is not suitable for publication in PLOS ONE.

• The content of paper is not enough for publication in PLOS ONE

• It is not observed new finding in this research.

Response: This study contributes to the existing body of knowledge on the epidemiology of occupational injuries in Ghana among workers from different occupational background. Particularly, Ghana is in the process of coming up with a single health and safety policy to promote health and safety at the workplace. This study provides useful information that can inform policy direction. 

Secondly, no study has been conducted in a major industrial city in the West African subregion, particularly Ghana, that estimates injury prevalence and determinants. This is a gap considering that the type of hazards that a worker in a non-industrial setting is exposed to may be different. Hence, it is essential to estimate the burden of injury and their contributing factors. 

Reviewer 2

Comment: Page 3, line 61 “The global burden of occupational injuries is estimated to be 1.9 million deaths in 2019” is this data/figure includes occupational diseases OR occupational injuries only?

Response: It includes diseases and this has been acknowledged in the revised manuscript. It now reads:

The global burden of occupational injuries and diseases is estimated to be 1.9 million deaths in 2019. [Page 3, lines 60-61]

Comment: Pages 4, line 88-92 The statement is about the associated factors; as result, it is better to merge with the previous paragraph.

Response: This has been done as suggested. Thank you for your suggestion. [Pages 3,4, Lines 83-87]

Comment: Pages 4, line 93: “Apart from the few studies mentioned above,…” What your study added to these few studies?

Response: This has been provided in the revised manuscript.

Apart from the few studies mentioned above, little is known about the epidemiology of occupational injuries among workers in a major industrial city with over five hundred companies and has the largest Port in Ghana. These studies only focused on estimating injury prevalence among specific working populations such as general construction workers (24), small-scale gold miners (22), healthcare workers (26), solid waste collectors (25) and Emergency Medical Technicians (27). This study estimated injury prevalence and their determinants among workers in different occupational groups (such as Port and Harbour, Manufacturing Companies, Tourism and Hospitality, Pharmaceuticals, among others). Estimating injury rates for the different occupational groups will give us a snapshot of their burden to inform policy direction towards an appropriate preventive strategy. This will help to improve the health and safety of workers in their working environment in Ghana and beyond. Continually measuring injury prevalence and its associated factors among workers in an industrial city will help to strengthen the body of evidence that policy makers, health and safety managers and other key stakeholders need to guide occupational health and safety. [Page 4, Lines 92-105]

Comment: Pages 5, line 123 Why you use 1.2 as design effect? Why not 1.5 or 2? Could you support with credible source/evidence 

Response: Thank you very much for your careful and meticulous observation on our manuscript. We have accepted your insightful observation on the sample size estimation relating to the design effect; the use of a design effect of 1.2 instead of 1.5. We acknowledge that this was an error and the necessary corrections have been effected in the manuscript. We apologize and will make sure that due diligence will be carried out in our future submissions. [Page 5, Line 131]

Comment: Pages 6, line 150-152 “’Occupational injury’ was operationalized as all injuries sustained while performing a task at the workplace that resulted in a loss of at least 12 hours of productive time” How did you confirm the impact of injury (a loss of at least 12 hours of productive time)?

Response: We used the International Labour’s definition of occupational injury to classify an injury as an occupational-related. To avoid misclassification of the outcome, participants who reported an occupational injury were asked to describe how the injury occurred. This has been duly recognized in the revised manuscript. This now reads as;

’Occupational injury’ was operationalized as all injuries sustained while performing a task at the workplace as defined by the International Labour Organization (30). The study participants who reported sustaining an occupational injury were asked to describe how the injury occurred to avoid misclassification. [Page 6, Lines 159-162]

Comment: Pages 6, line 160-161 Are you used Chi square or Fisher’s exact tests analysis? Better to state what you already used

Response: We used either Chi-square or Fisher’s exact (for proportions with subgroupings <5) tests. This has been acknowledged in the revised manuscript 

Associations between predictor (such as age, sex, health and safety training etc.) and outcome (occupational injury) variables were explored using either Chi-square or Fisher’s exact (for proportions with subgroups <5) tests. [Page 7, Lines 178-180]

Comment: Pages 7, line 185-186 “The proportion of study participants that experienced an occupational injury in the last 12 months was 64.7%.” State the proportion of occupational injuries with 95% CI; the lower and upper confidence intervals

Response: This has been done as suggested. The section in reference now reads as;

The proportion of study participants that experienced an occupational injury in the last 12 months was 64.7% (95%CI: 0.60 – 0.69). [Page 9, Lines 217-218]

Comment: Pages 8, line 197-199 “Over 54.9% of them were supplied with PPE and 77.3% of the study participants indicated that they used PPE at the workplace” How the proportion of PPE use is much higher than the proportion of PPE supply (77.5% vs 54.9%)

Response: Once again, this was an error which has been corrected in the revised manuscript. We are very grateful for drawing our attention to it. The revised section now reads;

Over 22.7% (104) of the workers indicated that they used PPE at the workplace. Out of the 104 workers that indicated they used PPE, about 24.0% of them indicated they used it always at the workplace. [Page 10, Line 229]

Comment: Table 2 Pages 9, line 207-209 For the variable “use of PPE at work”; you reported as an about 355 (77.3) are not using PPE. However, in the variable “Frequency of PPE use at work” you characterize the frequency of use of PPE for 355 study participants. You are expected to characterize the 104 study participants who reported as they use PPE

Response: Response: This has been done as suggested. It now reads as; 

Out of the 104 participants that indicated they used PPE, about 24.0% of them indicated they used it always at the workplace. [Page 10, Line 229-230]

Comment: How this can be? It is not clear are not inline. Please re consider the presentation of “use of PPE at work and Frequency of PPE use at work”Pages 10, line 230-232

Response: We have defined how these variables were measured in methods of the revised manuscript. It reads as;

The usage of PPE was measured by asking participants whether they wore PPE at the workplace when performing a task with a ‘yes’ or ‘no’ response while the frequency of PPE usage was assessed by asking participants how often they wear the PPE when performing a task at the workplace with ‘always’ or ‘sometimes’ response. ‘Always’ means that the worker wears PPE anytime he or she is performing a task while ‘sometimes’ means that the worker only wears the PPE as and when he or she deemed it fit (i.e. not wearing it all times when performing a task at the workplace). [Page 6, Lines 152-159]

Comment: “Workers that were working at the Port and Harbour were about 4 times (AOR: 4.32; 95%CI: 0.14 – 16.37) more likely to experience occupational injury compared to those at Agriculture and Forestry” The CI contains the null value (1), therefore, how this variable was considered as statistically significant covariate? 

Response: Thank you for drawing attention to this. This has been corrected in the revised manuscript. It now reads;

…… port and harbour (AOR: 3.77, 95%CI: 1.70 – 8.39)… [Page 11, Line 243]

Comment: Pages 11&12, line 239-241 Have you checked the assumptions of Binary logistic regression before conducting analysis? The variable” Frequency of PPE use at work” is not collected from all study participant rather only from 104 study participants; who reported as they used PPE. As result, can it can be regressed against Occupational injury? 

In the logistic regression analysis table (table 4), the outcome variable (occupational injury) with the number of injured and non-injured study participants shall be presented in a column, next to the list of variables column. 

Response: Thank you for this insightful comment. We have re-performed analysis on the logistic regression analysis again by using a backward stepwise approach. We did not include the frequency of PPE use in the model. [Page 7, Lines 181-190]

Comment: Pages 13, line 291-293 “It is recommended that there should be reduced payment benefits to the affected worker if investigations reveal he or she was not in PPE at the time of injury.” It shall be supported with credible evidence. What if the organization didn’t supply adequate PPE and workers unable to purchase and use by their own?

Response: We acknowledge your comment. The statement has been removed from the revised manuscript

Comment: Overall comment: Please work to improve the quality of the writing throughout the manuscript

Response: This has been done as suggested. Thank you very much

Reviewer 3

Comment: Page 2 line 27, please paraphrase it.

Response: This has been done as suggested 

Comment: Page 2 lines 43-44: try to reanalyze and interpret the association of your outcome variable with the level of education.

Response: This has been done as suggested. It now reads as;

Tertiary education (AOR: 0.03, 95%CI: 0.01 – 0.10) and secondary education (AOR: 0.22, 95%CI: 0.07 – 0.74) were protective of occupational injuries compared to basic education. [Page 2, Lines 46-47]

Comment: Page 2 line 45: “Working at Port 45 and Harbour (AOR: 4.32; 95%CI: 0.14 – 16.37)” Does it show a significant association?

Response: This was an oversight and it has been corrected. 

…..working at Port and Harbour (AOR: 3.77, 95%CI: 1.70 – 8.39) [Page 2, Line 43]

Comment: The magnitude of the problem should be clearly stated in an inverted-triangular form (global to local perspective). The authors should clearly show the research gap that they want to fill with the present study. They should intensively review previous research conducted in the area and direct how they will generate new evidence with their study.

Response: This has been done as suggested, please

Comment: The study setting and area should be mentioned precisely, even with the relative and absolute location. Therefore, a map of the study area should be included

Response: We are very grateful to the reviewer for his comments on including a map of the study area in the manuscript. However, we prefer the current state of the manuscript without the map due to copyright issues.

Comment: Page 5 lines 118-119: “All workers who reported non-occupational related 119 injuries and …… were excluded.” Do you mean all included subjects were those who have an occupational injury (Prevalence =100%)???

Response: No please, we included all workers between 20 and 60 years. The statement portraying that we had 100% prevalence has been removed from the revised manuscript

Comment: Page 5, line 123: why do you use a design effect of 1.2, which is nearly 1.00? What is the need to use the design effect? You have used two-stage sampling. What do you think about the proper design effect to be used for two-stage sampling? Please try to address it.

Response: Thank you very much for your careful and meticulous observation on our manuscript. We have accepted your insightful observation on the sample size estimation relating to the design effect; the use of a design effect of 1.2 instead of 1.5. We acknowledge that this was an error and the necessary corrections have been effected in the manuscript. We apologize and will make sure that due diligence is carried out in our future submissions. [Page 5, Line 131]

Comment: If there is more than one industrial worker in a household, how do you select the respondents? What inclusion and exclusion criteria were used during sampling?

Response: We interviewed all industrial workers in a household, if they were more than one. This was stated in the manuscript. This is because all the workers may not be exposed to the same working conditions at their workplaces. the inclusion and exclusion criteria were workers that were 20 to 60 years and have been working in Tema for at least, one year and all workers in administrative positions (such as receptionists, accounting staff, secretaries) were excluded from the study. This has been acknowledged in the manuscript. [Page 5, Lines 124-127]

Comment: How to control the reported bias (in case, some respondents may respond as if they were injured at the workplace, while it actually occurred in non-occupational areas)?

Response: To reduce bias, participants who reported experiencing injuries were asked to describe how and where the injury occurred. This has been acknowledged in the revised manuscript. It now reads:

“To avoid misclassification of occupational injury, study participants who reported sustaining work-related injuries were asked to describe how and where the injury occurred. ”. [Page 6, Lines 159-162]

Comment: Why do you use the KOBO toolbox and ODK data collection tools in combination? Use of the one will be simple for the management of the data and training of data collectors.

Response: We have revised the statement to reflect that we used Kobo Collect for the data collection. [Page 6, Line 148]

Comment: Some of your variables, for instance, the use of PPE, should be assessed through observation, not using a questionnaire. It raises the validity question of your finding.

Response: We agree with the reviewer on this comment. However, we pretested the data collection instrument and had a similar result. We have acknowledged the use of self-report to assess use of PPE as a limitation in the revised manuscript. [Page 15, Lines 342-343].

Comment: Page 10, line 231: Please check your presentation regarding the association of level of education with occupational injuries.

Response: This has been done as suggested. Thank you. The revised section now reads as;

Tertiary education (AOR: 0.03, 95%CI: 0.01 – 0.10) and secondary education (AOR: 0.22, 95%CI: 0.07 – 0.74) were protective of occupational injuries compared to basic education. [Page 11, Lines 246-347]

Comment: Page 11: On tabular presentation of the logistic regression,

- You should include the frequency and percentage of study participants per the dichotomous outcome variable (injured/non-injured). - The P-value for the crude odds ratio is not important

Response: Please, this was presented in Table 1 that is why we did not include it in the logistic regression table. We prefer to maintain the current state of how we presented the results in the tables. Thank you

Comment: Page 12, line 249: The prevalence is much higher than 50%, therefore, the phrase “more than half” is not descriptive of your finding.

Response: This has been revised to read… The prevalence of occupational injuries (64.7%) observed in the present study was high. [Page 13, Line 262]

Comment: The discussion and conclusions are well written

Response: Thank you very much.

---

## [Decision Letter · Decision Letter 1]

28 Jul 2023

PONE-D-23-10198R1Epidemiology of occupational injuries in an industrial city in Ghana: A cross-sectional surveyPLOS ONE

Dear Dr. Opoku,

Thank you for submitting your manuscript to PLOS ONE. After careful consideration, we feel that it has merit but does not fully meet PLOS ONE’s publication criteria as it currently stands. Therefore, we invite you to submit a revised version of the manuscript that addresses the points raised during the review process.

We look forward to receiving your revised manuscript.

Kind regards,

Aiggan Tamene

Academic Editor

PLOS ONE

Reviewers' comments:

Reviewer's Responses to Questions

**Comments to the Author**

1. If the authors have adequately addressed your comments raised in a previous round of review and you feel that this manuscript is now acceptable for publication, you may indicate that here to bypass the “Comments to the Author” section, enter your conflict of interest statement in the “Confidential to Editor” section, and submit your "Accept" recommendation.

Reviewer #4: (No Response)

Reviewer #5: (No Response)

Reviewer #6: All comments have been addressed

2. Is the manuscript technically sound, and do the data support the conclusions?

Reviewer #4: Yes

Reviewer #5: Partly

Reviewer #6: Partly

3. Has the statistical analysis been performed appropriately and rigorously? 

Reviewer #4: Yes

Reviewer #5: Yes

Reviewer #6: No

4. Have the authors made all data underlying the findings in their manuscript fully available?

Reviewer #4: Yes

Reviewer #5: Yes

Reviewer #6: Yes

5. Is the manuscript presented in an intelligible fashion and written in standard English?

Reviewer #4: No

Reviewer #5: Yes

Reviewer #6: No

6. Review Comments to the Author

Reviewer #4: a. basically, the writing is very good, but Should be given a year limit, whether the last 10 years or the last 5, so that the prevalence data is more relevant. It just needs to be added to the background, previous research related.

b. I applaud the authors for advancing understanding and emphasizing important risk about occupational injuries

c. As mentioned in the article, understanding this subject could provide an opportunity to tackle an important public health issue in the country, especially ini occupational injuries.

d. Please also provide the search build document you created with the identified number of articles at each library/data base.

e. Please make sure that all tables are mentioned in the main text and correctly numbered.

f. You indicated that characteristics including sex, level of education, industry type, and occupational injuries were pooled. Though I was unable to locate the studies that were combined to create those characteristics. Please mention them.

Reviewer #5: (No Response)

Reviewer #6: This is a cross-sectional survey of occupational injuries by industry in Ghana in 459 workers. Occupational injury rates were high, but the seriousness of these injuries were not well-described. The findings were not novel. Training programs are assumed to be important to preventing injury, but the evidence supporting this is generally weak. It is unclear what is meant by “safety creation awareness” in the abstract, but this might refer to safety climate or safety culture. Education levels have been shown to be important and we assume that those with higher education better understand the importance of following safety guidelines, but education could also be capturing language barriers. It is also unclear what “contract engagement” means. The problem of high occupational injury rates in LMIC is a problem that needs attention.

1. In terms of PPE use, is there evidence that the PPE provided to workers was functionally sound and adequate to the task? How often was it replaced? If it was worn and leaky or damaged, it would not be protective.

2. One of the most common questions asked injured workers is how much time was lost from work because this informs the seriousness of the injury. Was there a reason this question was not asked?

3. Why was sex, PPE, and monthly income not included in the multivariable model in Table 4? It was significant in the bivariate model. It seems that it would be important to control for sex and income in the model of injuries.

4. It does not seem as if PPE would be of great value in preventing the types of injuries described in Table 2, with the exception of possibly eye injury and burns if they were to the hands. Do you think PPE is acting as a surrogate for those who are more safety-minded?

5. The discussion does not make a case for specific regulations or policies that could be developed from the results of this study. How likely would it be to enforce policies around required safety and health trainings or use of PPE?

7. PLOS authors have the option to publish the peer review history of their article (what does this mean?). If published, this will include your full peer review and any attached files.

Reviewer #4: No

Reviewer #5: **Yes: **Mitiku Bonsa Debela

Reviewer #6: No

---

## [Author Response · Author response to Decision Letter 1]

15 Sep 2023

Reviewer 4 

Comment: Basically, the writing is very good, but Should be given a year limit, whether the last 10 years or the last 5, so that the prevalence data is more relevant. It just needs to be added to the background, previous research related.

I applaud the authors for advancing understanding and emphasizing important risk about occupational injuries

As mentioned in the article, understanding this subject could provide an opportunity to tackle an important public health issue in the country, especially ini occupational injuries

Response: We appreciate the comments of the reviewer 

Comment: Please also provide the search build document you created with the identified number of articles at each library/data base.

Response: This is going to be extremely difficult to produce precisely because we did not set out to conduct a narrative or scoping or systematic review. This manuscript was drafted by some of the authors and the others reviewed it for critical content in addition to providing statistical support. Thus, the authors are not in a position to provide a search build document as none was deliberately created.

Comment: Please make sure that all tables are mentioned in the main text and correctly numbered.

Response: All the tables have been mentioned in the text as suggested

Comment: You indicated that characteristics including sex, level of education, industry type, and occupational injuries were pooled. Though I was unable to locate the studies that were combined to create those characteristics. Please mention them.

Response: This was a primary study and did not involve pooling of any previous data. However, we provided information on how the questionnaire was developed to collect the study data.

The questionnaire was self-developed entirely new by the investigators through a literature review (10,21,23,33–37). Page 6, Line 164

Reviewer 5 

Comment: This is a cross-sectional survey of occupational injuries by industry in Ghana in 459 workers. Occupational injury rates were high, but the seriousness of these injuries were not well-described. The findings were not novel. Training programs are assumed to be important to preventing injury, but the evidence supporting this is generally weak.

Response: We agree with the reviewer on his comments about the seriousness of the injury not described in the manuscript. Collecting and reporting data on the number of hours lost from work due to the injury would have given a better view. We have therefore acknowledged it as a study limitation in the revised manuscript. 

The revised section now reads;

Again, this study did not report on how much time was lost from work due to the injury sustained at the workplace. Page 18, Line 420-421

Comment: Education levels have been shown to be important and we assume that those with higher education better understand the importance of following safety guidelines, but education could also be capturing language barriers.

Response: This has been captured in the discussion. The revised section now reads;

………Education could also be capturing some level of language barrier especially when most of these safety signs and guidelines are written in English which may cause a challenge in communication. Page 15, Lines 348-350

Comment: It is also unclear what “contract engagement” means. The problem of high occupational injury rates in LMIC is a problem that needs attention.

Response: They are casual staff and this has been replaced with ‘casual staff’ in the revised manuscript 

Comment: It is unclear what is meant by “safety creation awareness” in the abstract, but this might refer to safety climate or safety culture.

Response: This has been revised to read;

Key preventive measures to improve safety climate at the workplace can include training in health and safety, particularly, among contract staff and those with basic or no formal education

Page 2, Line 49

Comment: In terms of PPE use, is there evidence that the PPE provided to workers was functionally sound and adequate to the task? How often was it replaced? If it was worn and leaky or damaged, it would not be protective

Response: We did not investigate that so we are unable to report on them.

Comment: One of the most common questions asked injured workers is how much time was lost from work because this informs the seriousness of the injury. Was there a reason this question was not asked?

Response: was not asked and we have acknowledged it as a study limitation in the revised manuscript. The revised section now reads;

Again, this study did not report on how much time was lost from work due to the injury sustained at the workplace.

Page 18, Line 420-421

Comment: Why was sex, PPE, and monthly income not included in the multivariable model in Table 4? It was significant in the bivariate model. It seems that it would be important to control for sex and income in the model of injuries.

Response: We have included them in the table. However, sex was included in the variables for the stepwise multivariate logistic regress. However, the Stata output did not return the variable ‘sex’ when adjusted for with other covariates, because it was not deemed important per our data………..likely to be the case if there was an obvious majority of males

Table 4, page 12

Comment: It does not seem as if PPE would be of great value in preventing the types of injuries described in Table 2, with the exception of possibly eye injury and burns if they were to the hands. Do you think PPE is acting as a surrogate for those who are more safety-minded?

Response: The use of PPE is crucial in injury prevention especially in situations where both administrative and engineering controls are not feasible. After adjusting for significant variables in the stepwise logistic regression, PPE use did not have any significant effect on the occurrence of injuries.

Comment: The discussion does not make a case for specific regulations or policies that could be developed from the results of this study. How likely would it be to enforce policies around required safety and health trainings or use of PPE?

Response: We have revised the discussion to include advocating for training and retraining of workers on health and safety at the workplace. Again, we have also advocated for the Ministry of Trade and Industry to begin taking keen interest in occupational safety. They could for instance, have an overarching unit that visits industries regularly to ascertain adherence to industry standards

Page 18, Lines 444 – 455

Reviewer 6

Comment: First and foremost, I'd like to thank the authors of this research for their contribution to assessing Epidemiology of occupational injuries in an industrial city in Ghana: A cross-sectional survey. Next, I am grateful for the request to review the manuscript. In order to improve the readability of the manuscript the following points need further attention of the Authors:

Response: Thank you and well noted. We have addressed the manuscript based on your recommendations.

Comment: The title should be precise and clear. What is epidemiology for you? You said: Epidemiology of occupational injuries in an industrial city in Ghana. Classically epidemiology is defined as: The Study (interpreted to mean observation, recording, testing), Distribution (Pattern, Trends), Determinants (causes, risk factors) of health related states (incidence, death, treatment reaction, health care provision) and events (not just diseases), And Application of this study to detect, prevent & promotion of health in specified populations. So, modification of the title should be suggested as Prevalence and associated factors of occupation-related injuries in an industrial city in Ghana. 

Response: We are very grateful to the reviewer on his comments about the study title. The title of the study has been revised based on the reviewer’s comment.

The study title now reads as;

Prevalence and associated factors of occupational injuries in an industrial city in Ghana

Page 1, Line 1

Comment: The burdens and risk factors of occupation-related injuries in many occupational settings have been thoroughly established worldwide. So, what distinguishes this study from earlier research?

Response: It is true that occupational injuries have been studied across the globe. However, in Ghana there is limited data on occupational injuries. particularly, in the Tema Metropolis, the biggest industrial setting in Ghana, no study has estimated occupational injuries. Again, this study cuts across different industries unlike the others (22, 24-26) that focused on particular industries. Estimating injury prevalence in this setting is crucial for setting priorities in promoting employees health and safety in the country

Comment: Additionally, there were systematic and meta-analysis findings in the African region. Justify why a primary study is needed.

Response: This is true but the systematic review and meta-analysis did not include studies conducted in Ghana. This does NOT invalidate their findings but context-specific data are needed. Again, the particular context of Tema being an industrial hub in Ghana with limited studies relating to occupational injuries there need specific attention and hence the study.

Comment: The authors should indicate the types of research gaps. Attention to the population gap, methodological gap, knowledge gap, empirical gap, etc.

Response: This has been provided in the revised manuscript. 

Apart from the few studies mentioned, little is known about the prevalence and factors associated with occupational injuries among workers in a major industrial city with over five hundred companies and the largest Port in Ghana. These studies only focused on estimating injury prevalence among specific working populations such as general construction workers (24), small-scale gold miners (22), healthcare workers (26), solid waste collectors (25) and Emergency Medical Technicians (27) in Ghana. To the best of the authors knowledge, no study has estimated occupational injuries among workers from different occupations in the Tema Metropolis, the biggest industrial enclave in Ghana. Lack of evidence may affect the urgency to prioritize promoting employees’ health and safety in industrial cities in Ghana.

Page 4, Lines 100-107

Comment: The authors tried to demonstrate the need for the research in the last introduction sections of the revised manuscript, which is appreciated. However, the context and the need for the research should be elaborated for outsiders as well. Therefore, the authors should be expected to show the context and emphasize the novelty of the study. Scope of the research question addressed: The following factors could also play a significant role such as: The design of the working units, the structure of the working unit, and Ergonomic factors

Response: This has been provided in the revised manuscript. 

Apart from the few studies mentioned, little is known about the prevalence and factors associated with occupational injuries among workers in a major industrial city with over five hundred companies and the largest Port in Ghana. These studies only focused on estimating injury prevalence among specific working populations such as general construction workers (24), small-scale gold miners (22), healthcare workers (26), solid waste collectors (25) and Emergency Medical Technicians (27) in Ghana. To the best of the authors knowledge, no study has estimated occupational injuries among workers from different occupations in the Tema Metropolis, the biggest industrial enclave in Ghana. Lack of evidence may affect the urgency to prioritize promoting employees’ health and safety in industrial cities in Ghana.

Page 4, Lines 100-107

Comment: In the method section, it is good if the author(s) of this work ought to be addressing the following core issues:

Response: We have provided them as suggested please

Comment: the source population, study population, and study unit

Response: The target population for this study consisted of all workers who worked and were resident in Tema Metropolis in the Greater Accra Region of Ghana

Page 5, Lines 135-136

Comment: Inclusion and exclusion criteria: why peoples with age of 19 years old omitted? What is allowable age working in industry in Ghana? , justify why all workers in administrative positions (such as receptionists, accounting staff, and secretaries) were excluded from the study.

Response: We included workers age 18 to 60 years. however, we did not get any worker below 20 years. We have clarified that in the revised manuscript. Explanations have been provided for the exclusion criteria. 

The target population for this study consisted of all workers who worked and were resident in Tema Metropolis in the Greater Accra Region of Ghana. The selection of participants into the study was strictly based on satisfying the inclusion criteria which was being between 18 to 60 years (however, we did not get any worker below 20 years) old and having been working in Tema for at least twelve months. We excluded those 17 years and below because in Ghana, the minimum age of the engagement of a person that will expose him or her to hazard is 18 years (29). All workers in administrative positions (such as receptionists, accounting staff, and secretaries) were excluded from the study. These workers were excluded from the study because they were not directly involved in major activity at the workplace.

Page 5, Lines 135-142

Comment: Which sampling technique is appropriate for your study? Indicate sampling techniques with a simplified schematic diagram.

Comment: Two-stage sampling technique was used to select the study participants. This was provided in the original manuscript we submitted. We have provided the diagram as an Appendix and indicated in the manuscript. 

The revised section now reads; 

Participants were recruited based on a two-stage sampling procedure with probability proportional to size (Appendix 1). In low- and middle-income countries where it is difficult to accurately record individual households, this approach was used (35,36). In the first stage of the sampling, eight (8) communities (Communities 1, 2, 3, 4, 7, 8, 9, and 10) in Tema were randomly selected from twenty-five (25) communities through balloting. In the second stage, a simple random sampling technique was used to select households in each community using a random code generator. Households that had industrial workers staying there were identified and numbered in each community. In each community, a listing was done to obtain the number of households with industrial workers. In each selected household, all eligible participants who consented were interviewed. This method was repeated in each randomly selected household in all eight communities until the estimated sample size was obtained. Page 6, Lines 148-158

Comment: Data collection tools and techniques? Is the data collection techniques validated? If so show us the steps of tool validation.

Response: The data collection tool was not internationally validated and this has been acknowledged in the revised manuscript. And we have also outlined the steps that were taken to improve the validity of the tool

The questionnaire was self-developed entirely new by the investigators through a literature review (10,21,23,33–37) and has not been intentionally validated. However, the study adopted robust measures such as engagement of key stakeholders (epidemiologist, biostatistician, health and safety officer, language expects etc.) for both translation and back-translation of the questionnaire into the local Twi language and English respectively to improve its internal validity. Page 6, Lines 162-196

Comment: data quality assurance;

Response: This was provided in the original manuscript under data management and statistical analysis. In the revised manuscript, we have provided further clarity and presented it as a separate subheading “Data quality assurance.”

The revised section now reads;

Data quality assurance was conducted by the study statistician. The data was downloaded from the Kobo Collect in excel format and exported to Stata version 16.0 (StataCorp, College Station, USA) for quality management (data cleaning and coding) and analysis. The data was checked for completeness and consistency. Page 7, Lines 198-202

Again, we engaged research assistants who were final year postgraduate students who were knowledgeable in the subject area and they also underwent training on the data collection tool to improve. Page 7, Lines 183-189

Comment: Data analysis: attention to model fitness, the model predicting ability (sensitivity and specificity), multicollinearity etc

Response: This has been done as suggested.

A multicollinearity test was conducted to confirm the whether the explanatory variables that were included in the backward stepwise logistic regression had a correlation using the Variance Inflation Factor (VIF). The results indicated that there was no evidence of multicollinearity between the explanatory variables (mean VIF = 1.63, Maximum VIF = 2.13, Minimum VIF = 1.12) (See Appendix 1). Model fit was assessed using the Hosmer-Lemeshow test which showed that the model was good (p = 0.091). Page 8, Lines 192-224

Comment: How was the confounding variables addressed?

Response: We controlled for covariates that were insignificant but deemed important in the backward stepwise regression model (using a p-value of 0.1) Page 8, lines 213-215

Comment: How validity and reliability of the finding ensured?

Response: This was ensured by adopting an appropriate study design, and methods. The internal validity and reliability of the data collection tool were improved through pretesting and engagement of key stakeholders in developing the data collection tool as well as translation of the tool in the local Twi language for adequate comprehension on the part of all the study participants. 

The study adopted robust measures such as pre-testing, representative sampling (two-stage sampling), engagement of key stakeholders (epidemiologist, biostatistician, health and safety officers, language experts for both translation and back-translation of the questionnaire into the local Twi language and English respectively) in developing the questionnaire. This was done to improve its reliability and internal validity. 

Page 7, Lines 182-196 

Comment: What is the importance of adding a p-value to Table 1: Socio-demographic characteristics of study participants?

Response: This has been revised and the p-value has been removed from the table

Comment: Line 226-232: What were the dimensions of occupational health and safety measures investigated in this study? Hint: PPE utilization (as you stated), adherence to occupational safety commands, occupational health service utilization, workplace hygiene, and sanitation condition.

Response: We investigated PPE use and frequency of use, health and safety training, availability of health and safety department as well as worker’s satisfaction of health and safety measures at the workplace. This has been reported in Table 3 in the results section.

Comment: Your study was a community based how you evaluated utilization of PPE without conducting observation whiles the employees at job?

Response: We evaluated utilization of PPE through a self-report instead observational. This was captured under ‘Data collection’ in the original manuscript.

The usage of PPE was measured by asking participants whether they wore PPE at the workplace when performing a task with a ‘yes’ or ‘no’ response while the frequency of PPE usage was assessed by asking participants how often they wore the PPE when performing a task at the workplace with ‘always’ or ‘sometimes’ response. ‘Always’ meant that the worker wore PPE anytime he or she performed a task while ‘sometimes’ meant that the worker only wore the PPE as and when he or she deemed it fit (i.e. not wearing it every time a task was performed at the workplace).

Page 6, Lines 171-177

We are also aware of the limitation of adopting a self-report of PPE use and this was acknowledged as a study limitation under ‘Strength and limitation of the study’’. 

The approach of participant self-reporting of occupational injuries and the use of PPE could have affected the reported prevalence of occupational injuries and PPE use. Page 17, Lines 414-415

Comment: How were the overall percentages of occupational health and safety practices evaluated?

Response: We did not report on overall percentage of occupational health and safety practices. The focus of this study was to evaluate individual health and safety practices such as training, and use of PPE

Comment: Table 3: How health and safety measures satisfaction was checked at the workplace?

Response: This was checked by asking the study participants their overall satisfaction with health and safety measures at the workplace. This information has been provided in the data collection section of the revised manuscript. 

The revised section now reads;

Data on study participants’ satisfaction with existing health and safety measures at the workplace was also collected. Satisfaction with health and safety measures at the workplace was evaluated by asking the study participants their overall satisfaction with these measures with a ‘yes’ (meaning satisfied) or ‘no’ (meaning not satisfied) response. Page 6, Lines 168-173

Comment: Line 250, attention to Table 4: associated factors of occupational injury. Saying Determinants of occupational injuries among study participants is not the appropriate word, as the design of the study was cross-sectional. I suggested factors associated with occupational injuries among study participants.

Response: The title of Table 4 has been revised based on the reviewer’s recommendation.

It now reads as; 

Factors associated with occupational injuries among study participants.

Page 12, Line 289

Comment: Why were some AORs much larger than CORs? For instances: port and harbor (AOR: 3.77, 95%CI: 244 1.70 – 8.39), no health and safety training (AOR: 2.18, 95%CI: 1.08 – 4.39) and dissatisfaction with health and safety measures. I need an explanation of it.

Response: An explanation has been given in the revised manuscript. 

Variables such as “monthly salary of 1000 – 1500 cedis”, “ports and harbour”, “pharmaceuticals” and “health and safety training” had their adjusted odds ratio in the final multivariate logistic regression model higher increased from their crude odds ratio estimates. It is possible that this may be affected by some missing data in the other variables that were considered for the final logistic regression model or a confounder. The results in this study, however, are comparable to earlier studies and offer useful information about the factors contributing to occupational injuries.

Page 18, Lines 421-425

Comment: The authors should also think of effect modification.

Response: We checked for effect modification using an interaction. There was evidence of interaction between PPE usage and gender. However, this had a minimal effect on the overall model. Page 8, Lines 217-218

Comment: The reference categories should be included in regression table, 4.

Response: The reference categories have been added as suggested

Comment: The author should compare their findings with comparable populations and settings.

Response: There were some difficulties getting comparable populations in Ghana and Africa as our study cut across many occupations instead of just one as is the case in previous studies. The closest in Africa was a study that was conducted among industrial workers in Ethiopia by Hunegnaw et al., 2021 which we compared the injury rate with in our study. Outside Africa, we compared our study with another study which was conducted in India by Sashidharan and Gopalakrishnan (2017). Page 14, Lines 323-327

Comment: In order to improve the scientific quality of this finding, the authors should indicate what applications are there for their research findings? For example, explain how your findings enhance the general understanding of the topic to extend the reach beyond what others have found and give examples of why the world needs that increased understanding on topic under investigation.

Response: This has been provided as suggested. 

Comment: How did you relate your study to the existing theories in the literature? You should know the existing/established theories on the subject matter & link your findings to previous theories (Whether they agree or not).

Comment: It could have been better if the following were also discussed: As an example, limitations could be related to Social desirability bias, Recall bias, selection bias, information bias, generalizability and transportability, variable selection, etc.

Response: The study had some limitations. The approach of participant self-reporting of occupational injuries and the use of PPE could have affected the reported prevalence of occupational injuries and PPE use. Again, the use of self-approach could also lead to the introduction of social desirability bias. There may have been an element of recall bias since study participants had to do a retrospective (past 12 months) report of experiencing an occupational injury. However, study participants were asked to give an account of how the injury occurred to minimize the introduction of recall bias. Again, this study did not report on how much time was lost from work due to the injury sustained at the workplace. Variables such as “monthly salary of 1000 – 1500 cedis”, “ports and harbour”, “pharmaceuticals” and “health and safety training” had their adjusted odds ratio in the final multivariate logistic regression model higher increased from their crude odds ratio estimates. This may be affected by some missing data in the other variables that were considered for the final logistic regression model or a confounder. The results in this study, however, are comparable to earlier studies and offer useful information about the factors contributing to occupational injuries. 

Despite Tema being the largest industrial hub in Ghana, data obtained from only one site limits the generalizability of the study findings. However, since the data were collected from eight communities in Tema, the study findings can be generalized to this population. Another limitation of this study was the inability to estimate the long-term injury risk (for instance, adverse health outcomes in the future from present-day inhalation of gases) of the study participants. The data collection instrument was not intentionally validated. However, pretesting, the adoption of a standard definition of occupational injury, and a simple random sampling approach were adopted to reduce biases. 

Page 17, Lines 414-434

Comment: Don’t restate the results. Just conclude based on your information. For instance, you said: There is a high prevalence (64.7%????) of occupational injuries among the study participants. The same is true for discussion part

Response: This has been done as suggested. The revised section now reads;

There is a high prevalence of occupational injuries among the study participants.

Comment: You said: Future studies can adopt a qualitative approach to explore the satisfaction of health and safety policies at the workplace. Do you mean a mixed-methods approach? If so, do you think employing a qualitative approach alone is a satisfactory way to address the issue of OHS satisfaction? So, make a sound, actionable, and conclusion-based recommendation.

Response: We have revised that statement to reflect the adoption of a mixed method approach in future studies. Both quantitative and qualitative data can be important for addressing issue of OHS. 

It is imperative for effective mitigation measures to be instituted by management of these occupations for this high-risk population. These include training and retraining in health and safety, education and awareness campaigns as well as policy enforcement measures among workers, especially, those with a low level of education and temporary staff. The present study provides useful information that can guide efforts in the promotion of health and safety among workers at the workplace. The National Labour Commission of Ghana should also conduct routine monitoring and evaluations at various institutions (especially those in industrial cities) to ensure that employees health and safety are protected at the workplace. Future studies can adopt a mixed method approach to explore the satisfaction of health and safety policies at the workplace and how these contribute to occupational injuries. Page 18

Comment: References: Check thoroughly!

Response: Done as suggested please

---

## [Decision Letter · Decision Letter 2]

22 Feb 2024

PONE-D-23-10198R2Prevalence and associated factors of occupational injuries in an industrial city in GhanaPLOS ONE

Dear Dr. Opoku,

Thank you for submitting your manuscript to PLOS ONE. After careful consideration, we feel that it has merit but does not fully meet PLOS ONE’s publication criteria as it currently stands. Therefore, we invite you to submit a revised version of the manuscript that addresses the points raised during the review process.

The manuscript has been evaluated by four reviewers, and they have mostly voiced minor concerns. Specifically, the reviewers mention a need to include references for some statements, and to please ensure that there is no data missing in the presented tables.

Could you please carefully revise the manuscript to address all comments raised?==============================

We look forward to receiving your revised manuscript.

Kind regards,

Johanna Pruller, Ph.D.

Staff Editor

PLOS ONE

Journal Requirements:

Reviewers' comments:

Reviewer's Responses to Questions

**Comments to the Author**

1. If the authors have adequately addressed your comments raised in a previous round of review and you feel that this manuscript is now acceptable for publication, you may indicate that here to bypass the “Comments to the Author” section, enter your conflict of interest statement in the “Confidential to Editor” section, and submit your "Accept" recommendation.

Reviewer #4: All comments have been addressed

Reviewer #5: All comments have been addressed

Reviewer #6: All comments have been addressed

Reviewer #7: (No Response)

2. Is the manuscript technically sound, and do the data support the conclusions?

Reviewer #4: Yes

Reviewer #5: Yes

Reviewer #6: Yes

Reviewer #7: Yes

3. Has the statistical analysis been performed appropriately and rigorously? 

Reviewer #4: Yes

Reviewer #5: Yes

Reviewer #6: Yes

Reviewer #7: Yes

4. Have the authors made all data underlying the findings in their manuscript fully available?

Reviewer #4: Yes

Reviewer #5: Yes

Reviewer #6: Yes

Reviewer #7: Yes

5. Is the manuscript presented in an intelligible fashion and written in standard English?

Reviewer #4: Yes

Reviewer #5: Yes

Reviewer #6: Yes

Reviewer #7: Yes

6. Review Comments to the Author

Reviewer #4: I applaud the authors for advancing understanding and emphasizing Prevalence and associated factors of occupational injuries in an industrial city in Ghana, but u must describe which industrial this research focus? manufacture industrial or?

Though, there are grammatical and editorial problems that need corrections

Reviewer #5: Dear author, thank you so much for your outstanding revisions. Honesty speaking the authors comprehensively addressed my comments.

Reviewer #6: The English usage is excellent and the paper is well-written. I have one remaining question and that is, why are there missing values in table 4. I am assuming there is a reason that age and gender are missing from the multivariable logistic regression results, but could not find an explanation. Could these not be estimated?

Reviewer #7: Abstract

Results

There is no demographic data presented here. It's difficult therefore to appreciate the results since there is no data on demography.

Conclusion

The conclusion is vaguely written and does not seem to be supported by the results.

Line 79: is it ‘former’?

Line 378-382- This seems to be mere speculation. Please back it up with references.

Line 400-401- This is mere speculation. Please back it up with references. It’s obvious that workers with low pay may be prone to occupational injuries as supported my many studies. Please get some concrete reasons backed by literature as to why you had these findings

The conclusion should be short and concise and focus on the major findings from the work. Line 442 to 455 should be deleted. These are recommendations that are not part of conclusion.

7. PLOS authors have the option to publish the peer review history of their article (what does this mean?). If published, this will include your full peer review and any attached files.

Reviewer #4: No

Reviewer #5: **Yes: **Mitiku Bonsa Debela

Reviewer #6: No

Reviewer #7: **Yes: **Christian Obirikorang

---

## [Author Response · Author response to Decision Letter 2]

24 Feb 2024

Reviewer #4

Reviewer’s comment: I applaud the authors for advancing understanding and emphasizing Prevalence and associated factors of occupational injuries in an industrial city in Ghana, but u must describe which industrial this research focus? manufacture industrial or?

Authors’ response: The study adopted a multi-industry approach. We acknowledge the specificity of recommendations that come with studying specific industries but there is also a need for an overarching view of industrial injuries and their risk factors. 

Reviewer’s comment: Though, there are grammatical and editorial problems that need corrections.

Authors’ response: All grammatical and editorial problems have been addressed in the revised manuscript as suggested by the reviewer.

Reviewer #6

Reviewer’s comment: The English usage is excellent and the paper is well-written. I have one remaining question and that is, why are there missing values in table 4. I am assuming there is a reason that age and gender are missing from the multivariable logistic regression results, but could not find an explanation. Could these not be estimated?

Authors’ response: The variables ‘age’ and ‘gender’ were included in the final multivariable logistic regression analysis, but they they were inherently dropped in the stepwise regression build up by STATA. 

Reviewer #7: Abstract 

Reviewer’s comment: There is no demographic data presented here. It's difficult therefore to appreciate the results since there is no data on demography.

Authors’ response: We have presented some demographic variables in the results of the abstract as suggested by the reviewer. 

The mean age of the workers was 33.9 (±6.8) years with a range of 21 – 53 while over 18.1% of them were working at the Port and Harbour

Reviewer’s comment: The conclusion is vaguely written and does not seem to be supported by the results.

Authors’ response: The conclusion has been revised to reflect the results as suggested by the reviewer. The revised section now reads as;

The prevalence of occupational injuries in this study was high. Promoting machine tools' safety, health and safety training, and satisfaction with health and safety measures through rewarding workers who do not sustain injuries could be key to employees' health and safety.

Reviewer’s comment: Line 79: is it ‘former’?

Authors’ response: This is an error. it is ‘former.’ This has been corrected in the revised manuscript 

…who rely on the former to take action

Reviewer’s comment: Line 378-382- This seems to be mere speculation. Please back it up with references.

Authors’ response: The statement has been revised with a reference provided. It now reads as;

Casual workers are mostly inexperienced, less knowledgeable about occupational hazards and usually do not benefit from health and safety training at the workplace, predisposing them to occupational injuries (55,56)

Reviewer’s comment: Line 400-401- This is mere speculation. Please back it up with references. It’s obvious that workers with low pay may be prone to occupational injuries as supported my many studies. Please get some concrete reasons backed by literature as to why you had these findings

Authors’ response: We appreciate the comment made by the reviewer. The statement in reference has been taken off the revised manuscript. 

Reviewer’s comment: The conclusion should be short and concise and focus on the major findings from the work. Line 442 to 455 should be deleted. These are recommendations that are not part of conclusion.

Authors’ response: The conclusion has been revised and Line 442 to 445 has been deleted as suggested by the reviewer. 

There is a high prevalence of occupational injuries among the study participants and these were mainly caused by working tool and hot surface, substance or chemical. Factors such as employees’ level of education, type of engagement, monthly income, type of industrial work, health and safety training and satisfaction with health and safety measures at the workplace were independently associated with occupational injuries. The Ministry of Trade and Industry, Department of Factories Inspectorate, and National Labour Commission of Ghana should collaborate and conduct routine monitoring and evaluations at various institutions (especially those in industrial cities) to ensure that employees' health and safety are protected at the workplace. We also recommend that the Association of Ghana Industries (AGI) should create more awareness of employees’ health and safety by admonishing their members to be conscious of the need to adhere to safety practices at the workplace. Future studies can adopt a mixed-method approach to explore the satisfaction of health and safety policies at the workplace and how these contribute to occupational injuries.

---

## [Decision Letter · Decision Letter 3]

14 Mar 2024

Prevalence and associated factors of occupational injuries in an industrial city in Ghana

PONE-D-23-10198R3

Dear Dr. Opoku,

We’re pleased to inform you that your manuscript has been judged scientifically suitable for publication and will be formally accepted for publication once it meets all outstanding technical requirements.

Kind regards,

Emmanuel Timmy Donkoh, PhD

Academic Editor

PLOS ONE

Reviewers' comments:

Reviewer's Responses to Questions

**Comments to the Author**

1. If the authors have adequately addressed your comments raised in a previous round of review and you feel that this manuscript is now acceptable for publication, you may indicate that here to bypass the “Comments to the Author” section, enter your conflict of interest statement in the “Confidential to Editor” section, and submit your "Accept" recommendation.

Reviewer #5: All comments have been addressed

Reviewer #6: All comments have been addressed

Reviewer #7: All comments have been addressed

2. Is the manuscript technically sound, and do the data support the conclusions?

Reviewer #5: Yes

Reviewer #6: Yes

Reviewer #7: Yes

3. Has the statistical analysis been performed appropriately and rigorously? 

Reviewer #5: Yes

Reviewer #6: Yes

Reviewer #7: Yes

4. Have the authors made all data underlying the findings in their manuscript fully available?

Reviewer #5: Yes

Reviewer #6: Yes

Reviewer #7: Yes

5. Is the manuscript presented in an intelligible fashion and written in standard English?

Reviewer #5: Yes

Reviewer #6: Yes

Reviewer #7: Yes

6. Review Comments to the Author

Reviewer #5: (No Response)

Reviewer #6: The revised manuscripts addressed my concerns. I do wonder, however, if it would have altered the multivariable results to have forced gender into the models. There was a significant association with injury and gender in the univariable analysis. Probably, there are gender differences by industry which caused the gender variable to be dropped in backwards regression. Were the results confirmed using stepwise regression? It might be good to see how consistent the results are using a different model fitting algorithm.

Reviewer #7: All queries that were raised have been addressed. The manuscript has improved tremendously. It can the published in the current state

7. PLOS authors have the option to publish the peer review history of their article (what does this mean?). If published, this will include your full peer review and any attached files.

Reviewer #5: **Yes: **Mitiku Bonsa Debela

Reviewer #6: No

Reviewer #7: **Yes: **Christian Obirikorang

---

## [Editor Report · Acceptance letter]

20 Mar 2024

PONE-D-23-10198R3 

PLOS ONE

Dear Dr. Opoku, 

I'm pleased to inform you that your manuscript has been deemed suitable for publication in PLOS ONE. Congratulations! Your manuscript is now being handed over to our production team.

Kind regards, 

on behalf of

Dr. Emmanuel Timmy Donkoh 

Academic Editor

PLOS ONE